# An Investigation on the Effect of the Total Efficiency of Water and Air Used Together as a Working Fluid in the Photovoltaic Thermal Systems

**Mustafa Atmaca [1,*] and İmdat Zafer Pektemir [2]**

1    Faculty of Technology, Marmara University, 34722 İstanbul, Turkey
2    Institute of Pure and Applied Science, Marmara University, 34722 İstanbul, Turkey
*    Correspondence: matmaca@marmara.edu.tr

**Abstract:** The temperature of a PV (photovoltaic) panel increases when it produces electricity but its electrical efficiency decreases when the temperature increases. In addition, the electrical efficiency of the PV panel is very limited. To increase the PV efficiency, the rest of the solar irradiance must be used, together with the temperature being kept at an optimum value. With this purpose, an experimental study was conducted. Firstly, two specific photovoltaic-thermal (PV/T) systems were designed. The first was the PV/T system which used only a water heat exchanger. The other one was the PV/T system that used a water and air heat exchanger. In the latter PV/T system, air passed through both the top of the PV panel and the bottom of it while water passed through only the bottom of the panel in a separate heat exchanger. In this way, the water and air absorbed the thermal energy of the panel by means of separate heat exchangers, simultaneously. In addition to the two systems mentioned above, an uncooled photovoltaic module was also designed in order to compare the systems. As a result, three different modules were designed. This study was conducted in a natural ambient environment and on days which had different climatic conditions. The thermal, electrical and overall efficiencies of each PV/T module were determined. The results were compared with the uncooled module electrical efficiency. The results showed that when water and air were used together, it was more efficient than single usage in a PV/T system. The thermal gain of the working fluids was also found to be fairly high and so, the gained energy could be used for different purposes. For example, hot air could be used in drying systems and air condition systems. Hot water could be used in hot water supply systems.

**Keywords:** water- and air-based PV/T system; water based PV/T system; total efficiency; natural ambient conditions; solar energy

## 1. Introduction

Since the demand for energy is increasing day by day, renewable energy sources must be used more effectively, with ongoing research to find new renewable energy sources. The main energy source today is the sun. Therefore, solar energy is the most important kind of renewable energy, zero emission and zero waste being the main cause. The usage of solar energy consists of two main parts, the first, thermal solar energy and the second, electrical energy by means of photovoltaic panels.

The temperature while operating has an essential role in producing photovoltaic electric [1]. The temperature of a PV panel increases when it produces electricity. Therefore, the electrical efficiency of the panel decreases. The temperature of a PV panel should be at an optimum value. For this purpose, photovoltaic thermal systems weredesigned. These systems also enable us to obtain both thermal and electrical energy from the same panel. Water and air can be used as a working fluid in the

photovoltaic thermal systems. There are different specifications between air and water, in terms of physical properties. For example, water has got a corrosive character, but it is a higher thermal carrier than air. Air has a low thermal capacity but its corrosive character is less than water.

In the production of photovoltaic electricity, temperature has an essential role [1]. The electrical efficiency of a PV panel decreases due to the fact that its temperature increases when it starts producing electricity, which it should actually be at an optimum value. In regards to such a handicap, photovoltaic thermal systems were designed. These systems enable us to obtain both thermal and electrical energy from the same panel. In these photovoltaic thermal systems, water and air can be used as working fluids. However, in terms of physical properties, water and air have different specifications. For example, water has got a corrosive character, but is a higher thermal carrier than air. Air has a low thermal capacity but its corrosive character is less than water.

Sirinivas and Jarayaj [2] studied the PV/T system. They then emphasized that solar cells produce more electricity depending on higher solar irradiance. They also reported that the efficiency drops depends on a higher panel temperature. According to their study, the results showed that the production of electricity decreases with increasing temperature in the PV/T system.

Various studies were carried out to determine the PV panel temperature. One of these studies is that of Lasnier and Ang [3]. They presented the formulation below to determine pc-Si PV modules. ($T_{PV}$ is PV panel temperature, G is solar irradiance and $T_a$ is the ambient temperature in Equation (1))

$$T_{PV} = 30 + 0.0175(G - 300) + 1.14(T_a - 25). \tag{1}$$

According to Chandra et al. [4], the surrounding temperature causes a decrease in electrical efficiency because of the thermal loss. Whereas, wind has an important role in increasing the efficiency by reducing the temperature of the module. In another study, to determine the module temperature, Muzathik [5] presents the formulation below with an account of wind velocity.

$$T_{PV}(°C) = 0.943\text{Tambiant} + 0.0195\text{Irridance} - 1.528\text{Wind speed} + 0.3529. \tag{2}$$

Bardhi et al. [6], in their study about parameters that are effective on the PV panel temperature, conclude saying that the planning of the module, the mounting parameters, the electrical variables, irradiance, the temperature of the ambient, wind speed and wind direction, all influence the PV panel temperature.

Another important issue in the field of photovoltaic systems is that the efficiency decreases when the panel temperature increases. Kalogirou and Tripanagnostopoulos [7] studied the decrease of the PV panel electrical efficiency and pointed out that in the case of a single degree increase in the temperature, a decrease of 0.45% in efficiency occurs in monocrystalline and polycrystalline cells. The working temperature of a PV panel is effective on the electrical efficiency, while the performance of the PV panel depends on sufficient cooling [8].

The evaluations made above clearly show that to increase the electrical efficiency in photovoltaic modules cooling plays a major role. For such a cooling process, the thermal energy gained from the photovoltaic thermal systems is suitable to be used.

In their studies, Sopian et al. [9] used a double pass PV/T collector which was suitable for solar drying systems. As a result, they emphasized that photovoltaic collectors are suitable for many applications such as solar drying and photovoltaic (BIPV) systems which are integrated into buildings.

Ben Cheikh el hocine et al. [10], conducted a parametric study with galvanized iron of high quality and removed heat in the module by natural circulation of air. The thermal efficiency of the PV/ T collector was found to be about 54.51% in the mode of the water heat exchanger, and 16.24% in air heat extraction. The electrical efficiency was 11.12%, for a sample climatic, operating, and designed parameters.

Jouhara et al. [11] developed a new heat pipe design and conducted their study both by sticking the PV on this design and without having it stuck on it. They confirmed and tested their studies through a full-scale test in Cardiff, UK. According to the test results, the efficiency of the solar/thermal conversion was found as 64% for the collector with no PV and 50% for the collector with the PV.

In another study, Mojumder et al. [12] designed and fabricated a prototype PV/T collector with fins used for cooling. According to the results of the study, they obtained a maximum thermal efficiency of 56.19% and a maximum PV efficiency of 13.75% for the four fins at 0.14 kg/s of mass flow and 700 W/m$^2$ of solar irradiance.

Abu Bakar et al. [13] proposed a system integrated into the PV module that has a serpentine shaped-copper tube and a single pass air channel. They used air and water as working fluids. According to the simulation results, better performance was obtained when the fluids operated simultaneously.

Alobaid et al. [14], investigated the overall performance of the water based PV/T system using a mathematical model. According to their results, the electrical efficiency changed between 14.7% and 15.5% depending on the PV cell temperature. In addition, the thermal efficiency changed between 72% and 83% depending on several factors such as solar irradiance, ambient temperature and the inlet and outlet fluid temperature. In another recent study, Kazem [15] investigated the electrical performance of a specific water based PV/T system. In the PV/T system, he used stainless steel, a good thermal conductor. To improve the heat transfer area between the pipe and the collector he used a rectangular shape pipe. According to the results, the average PV/T panel power was 6% higher than the average power of the PV panel.

In this present study, we aimed to gain three different forms of energy by using the same solar panel. Since the purpose was not only to obtain effective cooling but also a good overall efficiency, more electrical power was not produced from the PV/T system than the PV module, as it was in the study of Kazem [15]. However, the overall efficiency was well high and close to the values in the study of Alobaid et al. [14]. Ben Cheikh el hocine et al. [10] used a natural air circulation in the PV/T system as the exchanger of air temperature and achieved 16.24% thermal efficiency. On the other hand, Mojumder et al. [12] used the cooling fins and they obtained a maximum thermal efficiency of 56.19%.

Jia et al. [16] summarized the different technologies of PV/T (photovoltaic-thermal) systems under different atmospheric conditions by using different working fluids. They investigated different types of flat plate PV/T systems, concentrator type PV/T systems besides a photovoltaic integrated thermoelectric cooler. In addition, they investigated which PV/T technology was suitable under what solar applications.

They made comparisons between different types of PV/T collectors that were used with different working fluids. According to their study, the air type PV/T system has a simple design while the water type has a complex structure.On the other hand, the air type PV/T has a low thermal performance but the water type PV/T has a higher thermal and electrical efficiency than the air type PV/T. Bifluid PV/T systems use water and air have a high thermal and electrical performance, enabling them to provide hot water and hot air simultaneously, and being better at cooling the photovoltaic panel but having a complex structure, high cost and limited applications.

According to the study, in the unglazed single pass air based PV/T system, which was studied by Bambrook and Sproul [17], the achieved thermal efficiency was 28%–55%; while the water based PV/T system studied by Zhao et al. [18] the achieved thermal efficiency was 40% and in the one studied by Chow et al. [19] was 45%–52%. On the other hand, in the air and water based bi-fluid system, studied by Jarimi et al. [20], the thermal efficiency was observed as 65.7% at fixed air flow and it was 66.12% at fixed water flow. However, while for the air cooling, a forced air circulation and specific honeycomb fins were used, copper tubes were used for the water cooling. The thermal efficiency was found to be between 50% and 85%. At last the three forms of energy were produced together with a satisfying level of overall performance.

Here we studied photovoltaic-thermal systems using both air and water, and its comparison with water based systems and the uncooled PV system. Our study was conducted as an experimental study

so, we first designed two specific PV/T systems. The first one was a PV/T system that used only a water heat exchanger. The second one was a PV/T system that used a water and an air heat exchanger. Using water and air together made it possible to benefit from their different physical properties. While air passed both the top of the panel and the bottom of it, the water passed only through the bottom of the panel in a separate heat exchanger. In this way, the water and air absorbed the thermal energy in the panel in separate exchangers at the same time. Thus, it cooled both the top and the bottom of the panel at the same time.

Having an uncooled photovoltaic panel to use for comparison in the experimental setup, three different modules were designed for the study. The study was conducted in different climatic conditions. The thermal efficiency besides the electrical and overall efficiencies of each module was determined and the results showed that the thermal gain of the working fluids was fairly high. This proves that it could be used for several purposes by optimization.

The comparisons between the water and air based PV/T with a water based PV/T and an uncooled PV was also included in the study, having been conducted under natural ambient conditions contrary to most common studies. The connection point of the serpentine, turbulator and black absorber plate was new. This new system enabled us to benefit from the solar energy by a proportion up to 60%–90%, making it convenient for buildings which need heating and hot water.

## 2. System Description

The experimental system consists of three main systems; the cooling system, electrical system and data system. The systems are described under the next section title and the main parts of the system are described in Section 2.1 below. In Section 2.2 the details of the cooling electrical and data systems are described.

### 2.1. Main Parts of the Experimental Set-Up

The main parts of the experimental set up are shown in Figure 1 and listed below.

1.  Glass (cover of the module-3): Distance: 4mm, transmittance: 0.92.
2.  PV Panel (same in all modules): TPSM6U Monocrystalline 200 W, $V_{oc}$:45.4 V, $I_{sc}$:5.77 A.
3.  Absorber plate (on the module-2 and module-3): Distance: 0.400 mm, material: aluminum ($\lambda$: 200 W/mK), painted black.
4.  Back plate (on the module-3): Distance: 2 mm, material: steel.
5.  Air fan (on the module-3): With electronically commutator.
6.  Water serpentine (on the module-2 and module-3): Collector's ext.diameter: 32 mm, tubes ext. diameter:10 mm, material: copper($\lambda$: 394 W/mK).
7.  Turbulator (on the back of module-3): Honeycomb, distance: 1.5 mm, length: 1580 mm, material: aluminum ($\lambda$: 200 W/mK).
8.  Pump: (on the module-2 and module-3): With frequent convertor.
9.  Boiler: For heat exchanging, capacity: 100 L, placed on the top level of the modules.

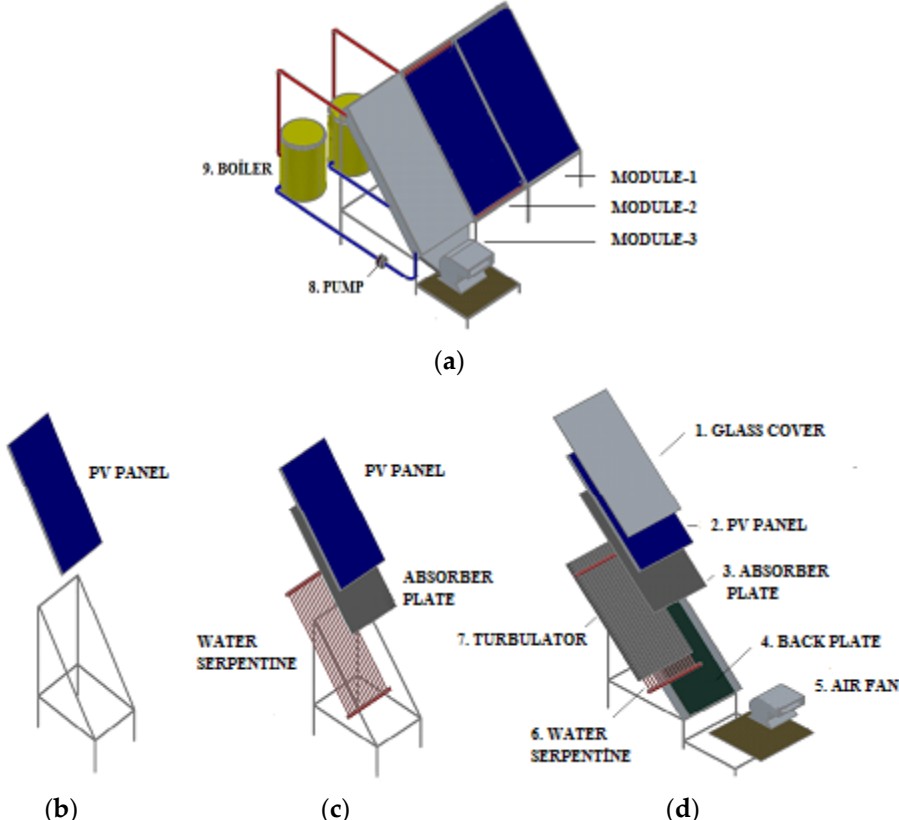

**Figure 1.** Description of the experimental set-up. (**a**) General shematic view of experimental set up. Module-1 is: uncooled photovoltaic module, Module-2 is: PV/T module that cooled by water, Module-3 is : PV/T module that cooled by water and air; (**b**) Module-1; (**c**) Module-2; (**d**) Module-3 [21].

### 2.2. Specifications of Cooling, Electrical and Data Systems

The location of the study was in the center of Konya Province in Turkey with the latitude 37°N and longitude 32°E. The tilt angle was decided as 33° for all seasons. Each of the three modules was mounted with the same tilt angle (In Figure 1). A mineral-based insulation was applied on module-2 and module-3 with a thickness of 50 mm and $\lambda$: 0.035 W/mK.

The absorber plate and copper serpentine were pasted with a paste material with a thermal conductivity of 0.88 W/mK. For the air turbulator configuration, a honeycomb fin was selected because of its large surface area (In Figure 2). The thermal efficiency of this model was emphasized in the literature. The Thermal efficiency of the collector with honeycomb fins was 87% and without the fins it was 27% at the 0.11 kg/s of mass flow and 828 W/m$^2$ of solar irradiance [22].

The placement of the modules is shown in Figure 3 and the placement of the sensors is shown in Figure 4. All input signals taken from the sensors were transferred into the industrial modules which were then converted into the RS 485 Modbus RTU protocol. Finally, the data were transferred to a computer.

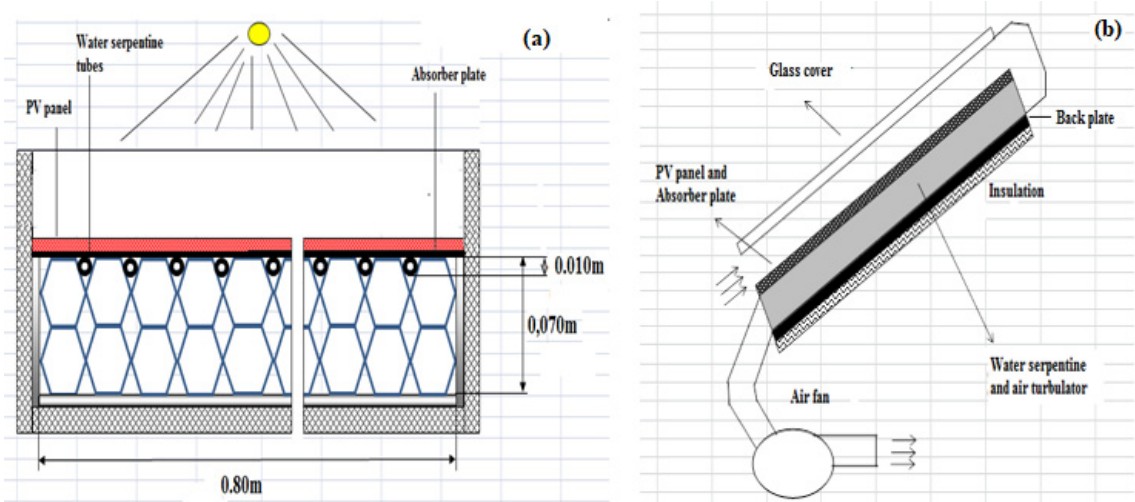

**Figure 2.** Details of module-3 [21] (**a**)cooling unit of module-3 (**b**) cross-section of module-3.

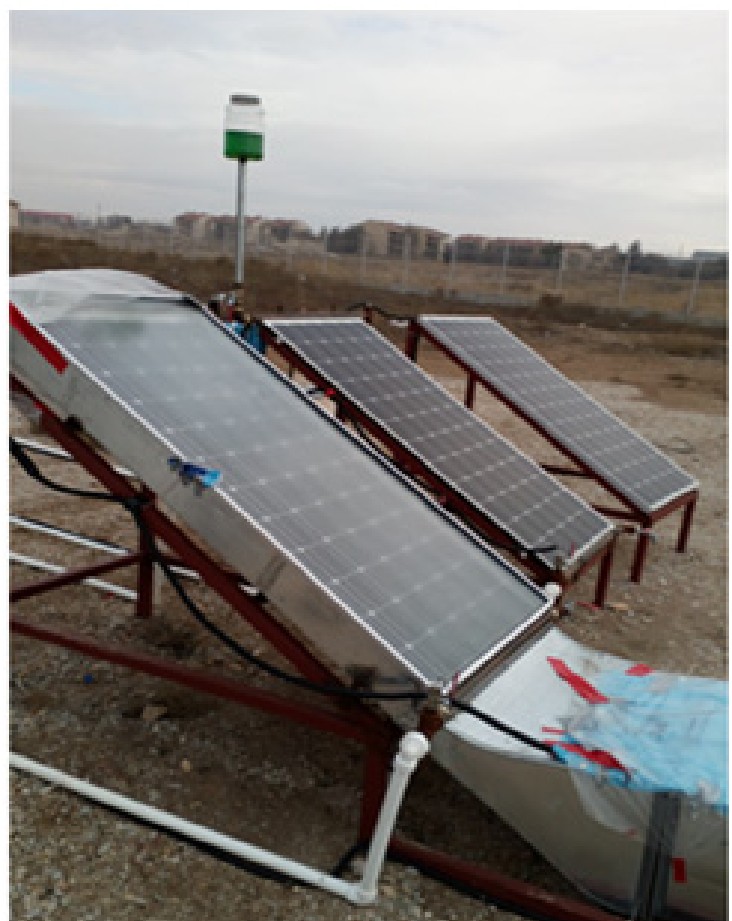

**Figure 3.** General view of experimental set up.

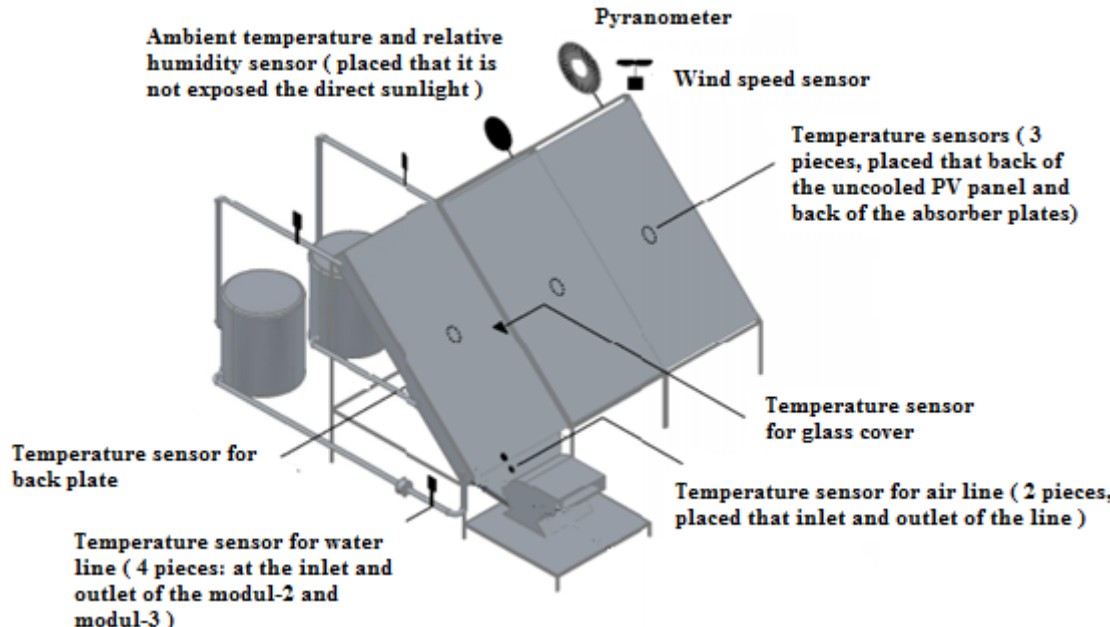

**Figure 4.** Placement of sensors.

The water and air circuit installation are shown in Figure 5 [21]. The water circuit installation of each module was the same. An ethylene glycol–water mix was used for the water circuit (in Figure 5) in case of a risk of freezing. The water ratio was 2/3 and the temperature was between 20 and 40 °C for the mixture. The specific heat ($C_p$) and density of the mixture were selected as nearly 0.885 kcal/kg°C (3699.3 j/kg°C) and nearly 1050 kg/m³ respectively by means of Duffie and Beckman's study [23].

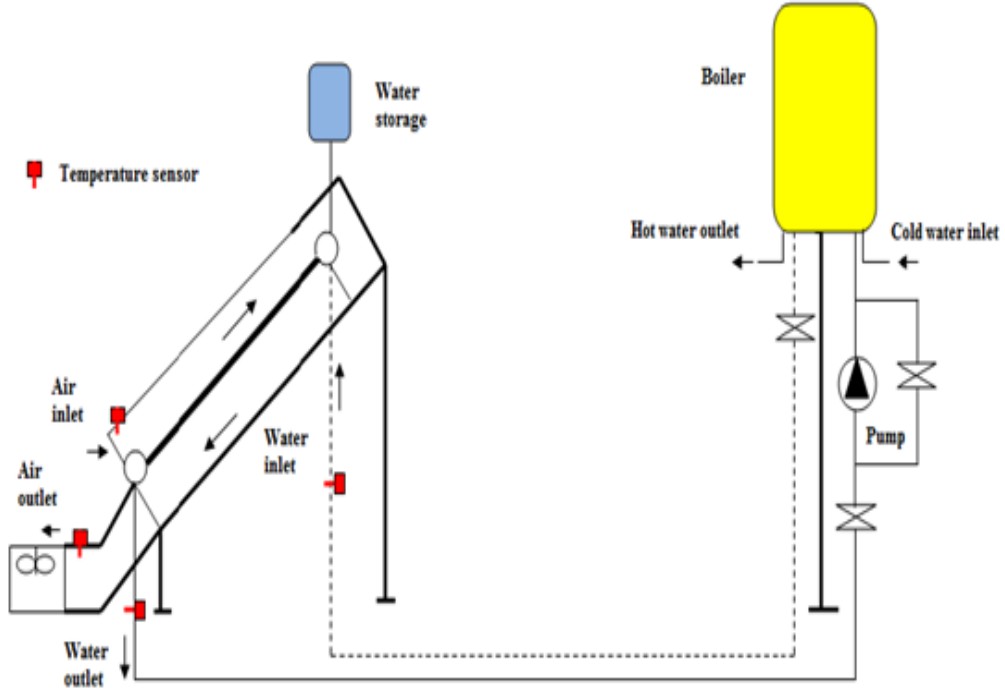

**Figure 5.** The air and water circuits of module-3 [21].

The PV module, load or demand, battery storage, controller and DC-AC inverter are part of a normal solar PV system [24]. In the present study, six accumulators were used for the electrical energy storage while 2 accumulators were connected to each PV module. A charge controller and an inverter were used for each PV module and an air fan and several pumps were connected to separate modules.

The performance of a PV generator changed the load, decreased with increased solar irradiance and increased temperature [25]. The open-circuit voltage and short circuit current of each module were measured by means of an electrical command box. Such a way of measuring is more precise to determine the PV electrical production capacity because the production of electrical energy affects the load device and accumulator capacity as well as the charge controller.

## 3. Analysis

### 3.1. Equations

The equations which were used to determine the performance of the different modules are mentioned below. These performance values include the powers of the modules and the efficiencies of the modules. The powers of the modules as electrical, thermal and total power, the efficiencies of the modules as electrical, thermal and overall efficiency.

$$\text{Electrical power: } I_{sc}V_{oc} \tag{3}$$

$$\text{Thermal power of module-2: } P_{th\text{-}m2} = \dot{m}_{water}C_{p(water)}(T_o - T_i) \tag{4}$$

$$\text{Thermal power of module-3: } P_{th\text{-}m3} = \dot{m}_{water}C_{p(water)}(T_o - T_i) + \dot{m}_{air}C_{p(air)}(T_o - T_i) \tag{5}$$

$$\text{Total power of module-2: } P_{tot\text{-}m2} = I_{sc}V_{oc} + \dot{m}_{water}C_{p(water)}(T_o - T_i) \tag{6}$$

$$\text{Total power of module-3: } P_{tot\text{-}m3} = I_{sc}V_{oc} + \dot{m}_{water}C_{p(water)}(T_o - T_i) + \dot{m}_{air}C_{p(air)}(T_o - T_i) \tag{7}$$

$$\text{Total irradiance: } I_{tot} = I_t A \tag{8}$$

Electrical efficiency:

$$\mu_e = \frac{I_{sc}V_{oc}}{I_t A} \tag{9}$$

Thermal efficiency of module-2:

$$\mu_{th-m2} = \frac{\dot{m}_{water}C_{p(water)}(T_o - T_i)}{I_t A} \tag{10}$$

Thermal efficiency of module-3:

$$\mu_{th-m3} = \frac{\dot{m}_{water}C_{p(water)}(T_o - T_i) + \dot{m}_{air}C_{p(air)}(T_o - T_i)}{I_t A} \tag{11}$$

Overall efficiency of module-2:

$$\mu_{o-m2} = \frac{\dot{m}_{water}C_{p(water)}(T_o - T_i) + I_{sc}V_{oc}}{I_t A} \tag{12}$$

Overall efficiency of module-3:

$$\mu_{o-m3} = \frac{\dot{m}_{water}C_{p(water)}(T_o - T_i) + \dot{m}_{air}C_{p(air)}(T_o - T_i) + I_{sc}V_{oc}}{I_t A}. \tag{13}$$

Water mass flow= it was calculated by means of total heat loss of the water based collector and average inlet and outlet temperature of the water.

$$\text{Air mass flow = Airspeed} \times \text{cross-area of air channel (airspeed was measured by an airspeed sensor on the outlet of the air channel)} \tag{14}$$



### 3.2. The Evaluation Values Compared for the Days that Have Different Climatic Conditions

3.2.1. Body Temperatures with Meteorological Conditions

The body temperature tendencies of the modules with meteorological conditions on 29.04.2018 are shown below in Figure 6. Cooling by water was initiated at 09:09 in the PV/T modules, while cooling by air was initiated at 12:00 in module-3. The ambient air was clear and windy. The module-1 and module-2 temperatures were almost the same but the temperature in module-3 being higher than the other modules.

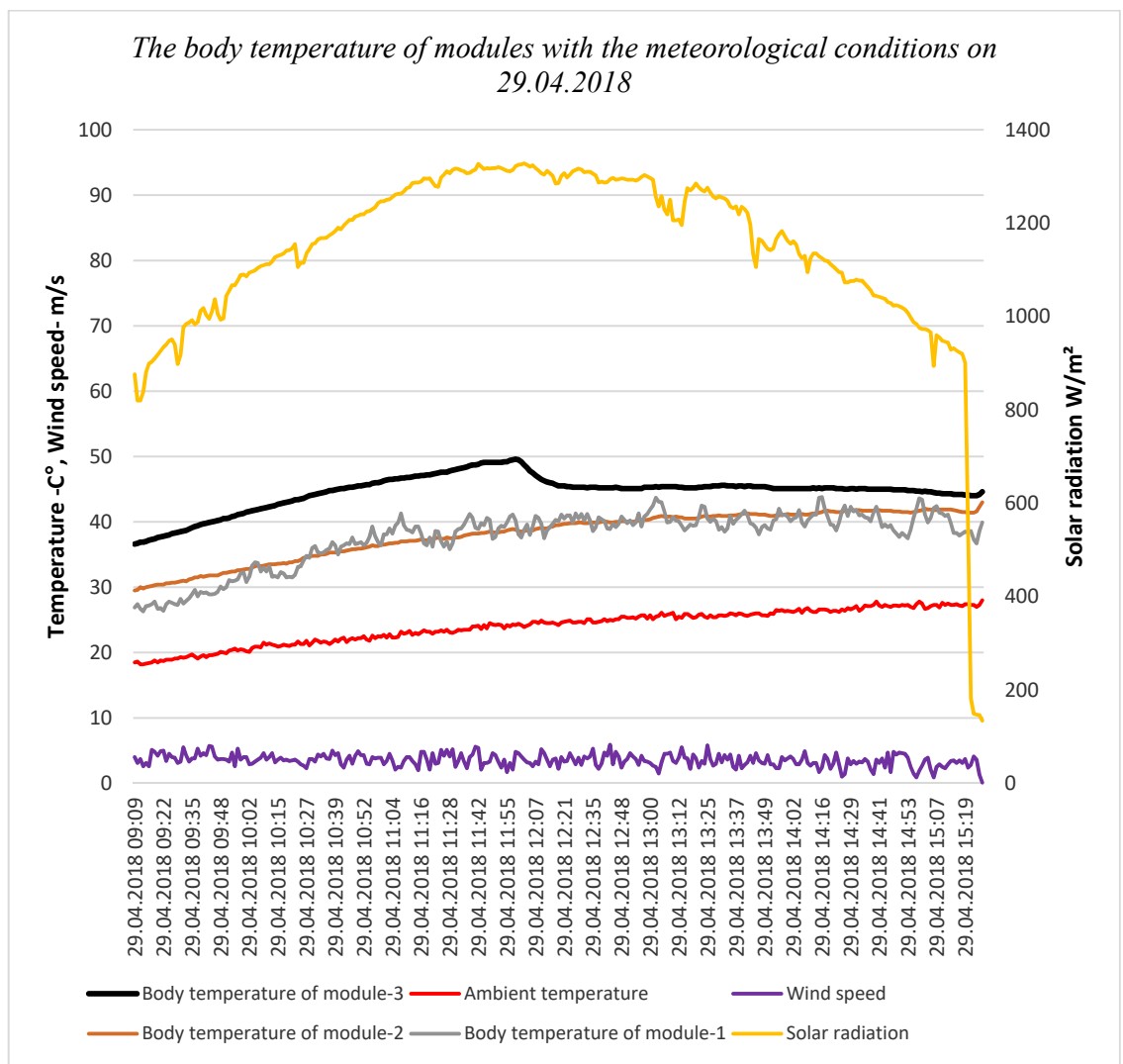

**Figure 6.** The body temperature of the modules with the meteorological conditions on 29.04.2018.

The body temperature tendencies of the modules with meteorological conditions on 01.07.2018 are shown below in Figure 7. Cooling by water and air were initiated at 12:00 in the PV/T modules. The ambient air was clear and calm. Therefore, the wind speed was not recorded. The temperatures in module-2 and module-3 were close to each other but the temperature in module-1 was higher than the other modules.

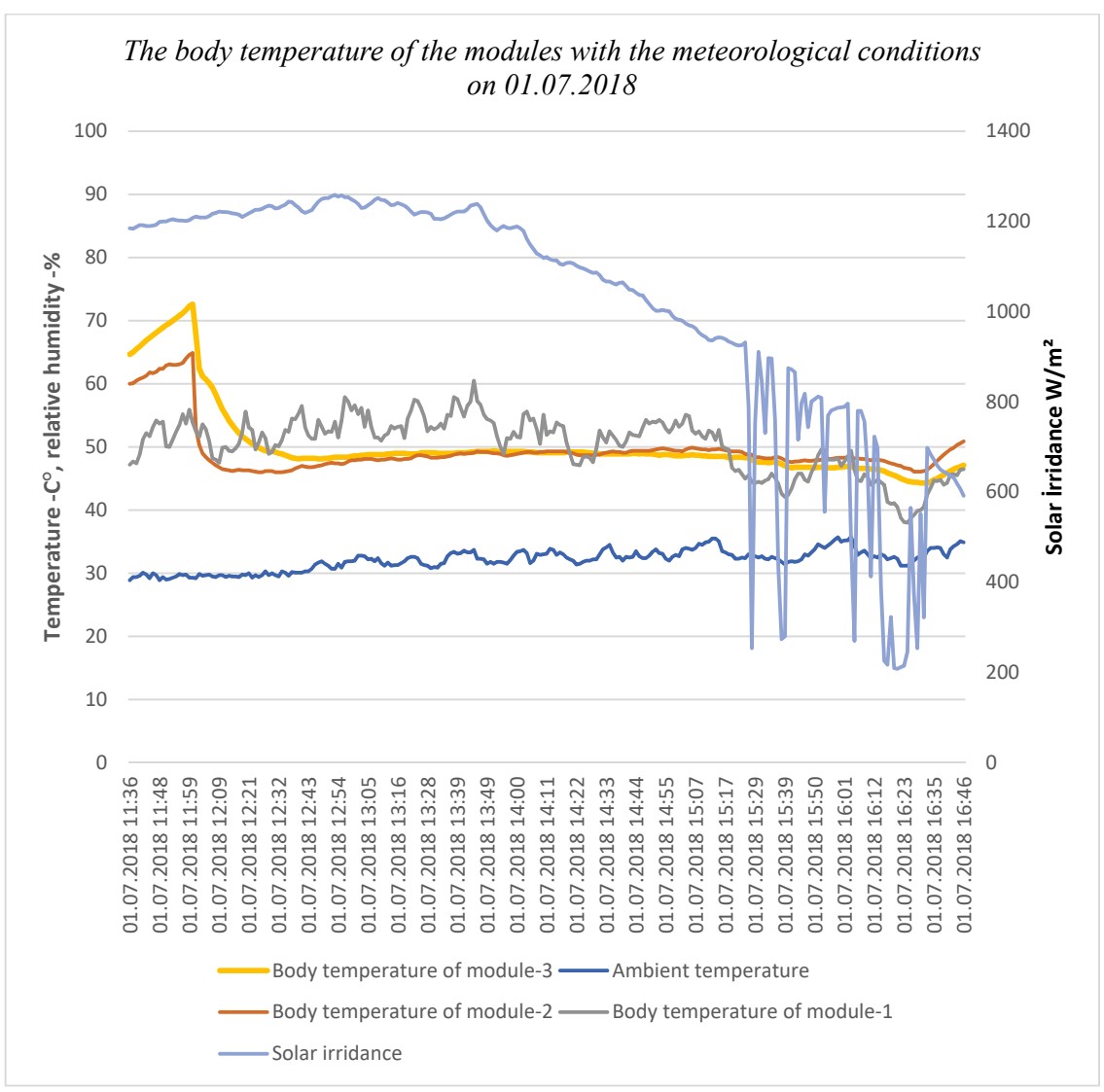

**Figure 7.** The body temperature of the modules with the meteorological conditions on 01.07.2018.

### 3.2.2. Powers and Efficiencies

We can observe in Figures 8–11 below that production of electrical power and the efficiencies show a similar trend with solar irradiation. The solar irradiance interval was nearly midday on 29.04.2009 (In Figure 6), but the solar irradiance trend was from midday to the end of the day on 01.07.2018 (In Figure 7). For this reason, electrical power and the efficiency showed a sharp decrease on 01.07.2018. In module-2, the electrical efficiency and the power on 01.07.2018 were closer to the efficiency of module-1 than the value on 29.04.2018 because the cooling was more effective on 01.07.2018 (in Figure 7).

We can see in Figures 12 and 13 that the thermal power in module-2 was between 200 and 300 W, but the thermal power in module-3 decreased to between 1400 W and 800 W because the gained air heat was more affected from the changing solar irradiance. The thermal efficiency in module-2 was nearly stable on 29.04.2018 (in Figure 14), but the thermal efficiency in module-2 and module-3 showed sharp waves (in Figure 15) because of the solar irradiance waves on 01.07.2018 (in Figure 7). On the other hand, the gain of air heat showed sharp increases and decreases because of the sharp changing of input air temperature and output air temperature.

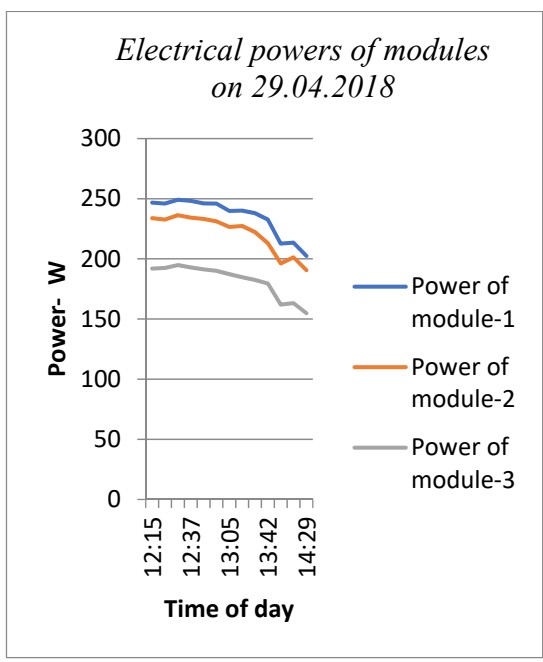

**Figure 8.** Electrical powers of modules on 29.04.2018.

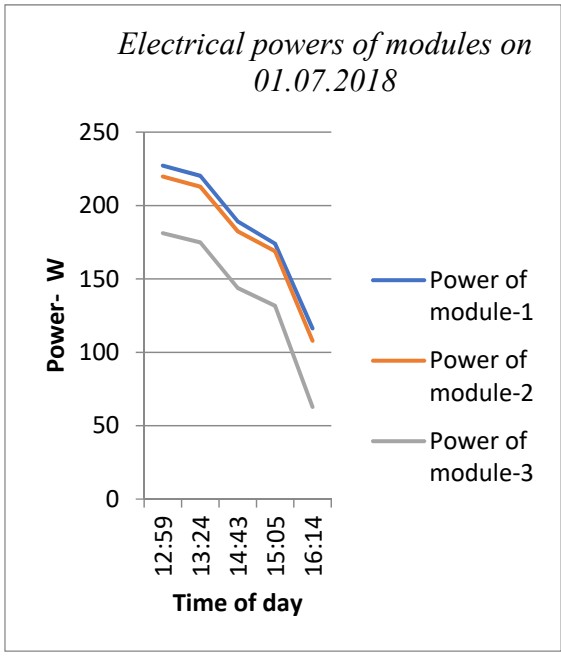

**Figure 9.** Electrical powers of moduleson 01.07.2018.

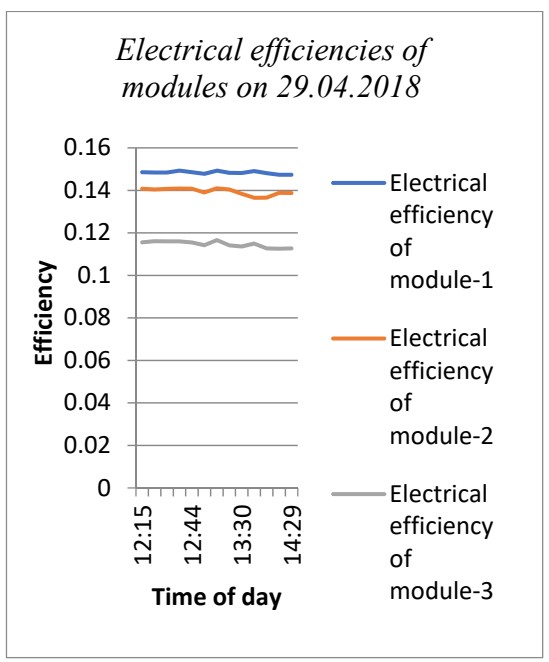

**Figure 10.** Electrical efficiencies of modules on 29.04.2018.

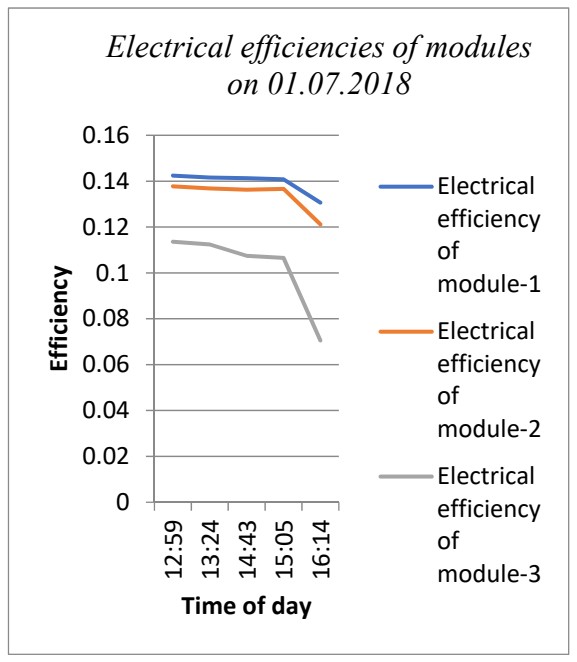

**Figure 11.** Electrical efficiencies ofmoduleson 01.07.2018.

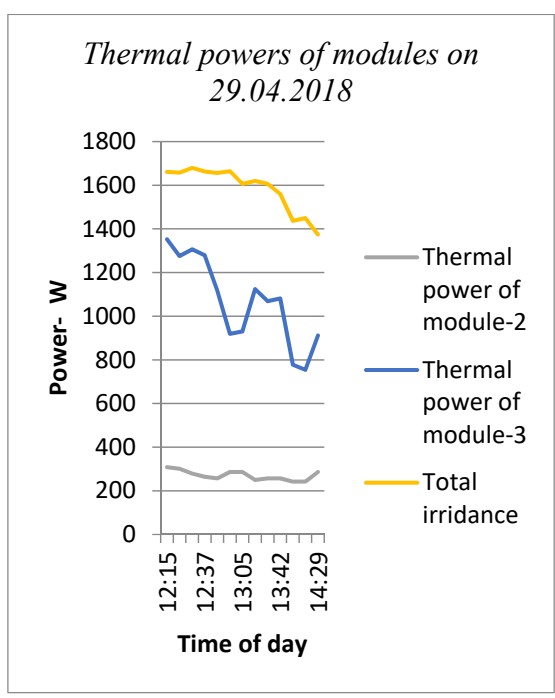

**Figure 12.** Thermalpowers of modules on 29.04.2018.

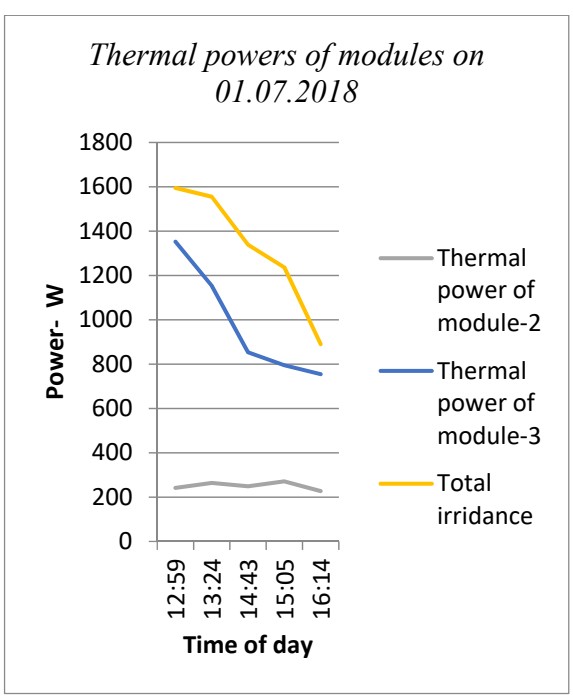

**Figure 13.** Thermal powers of modules on01.07.2018.

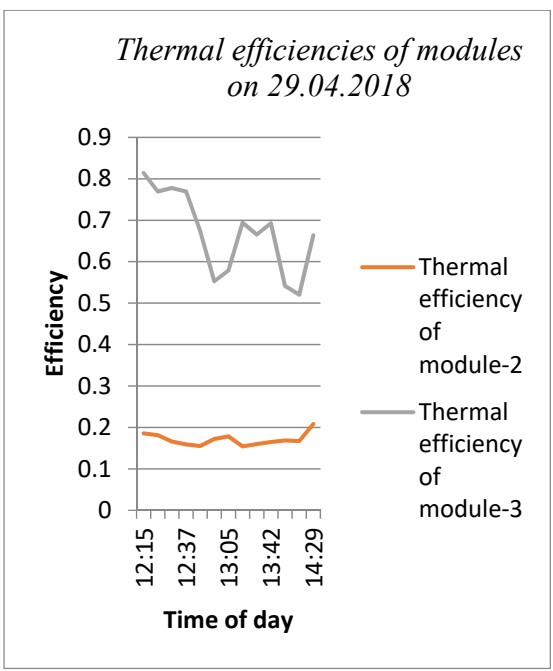

**Figure 14.** Thermal efficienciesof modules on 29.04.2018.

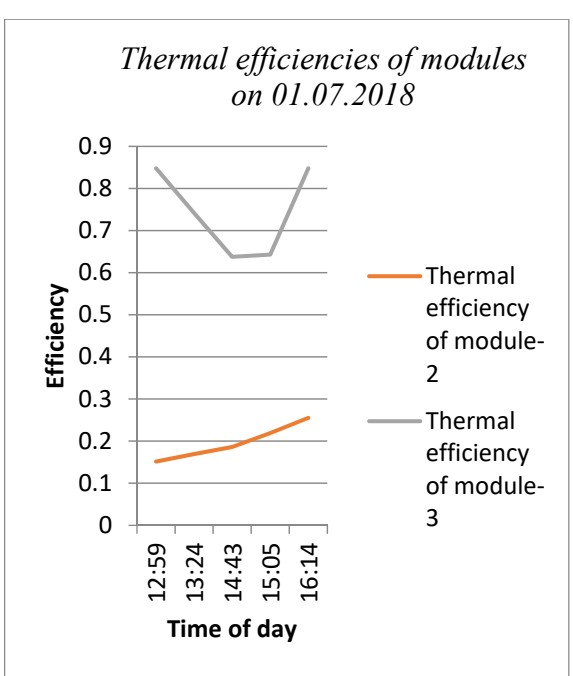

**Figure 15.** Thermal efficiencies of modules on 01.07.2018.

We can see the decreasing trend in the total power of the modules (in Figures 16 and 17). This situation was related to the selected time interval of the day. On each day, the selected time interval was from midday to the end of the day, but the decreasing trend was sharper on 01.07.2018 than on 29.04.2018 because of the selected time interval.

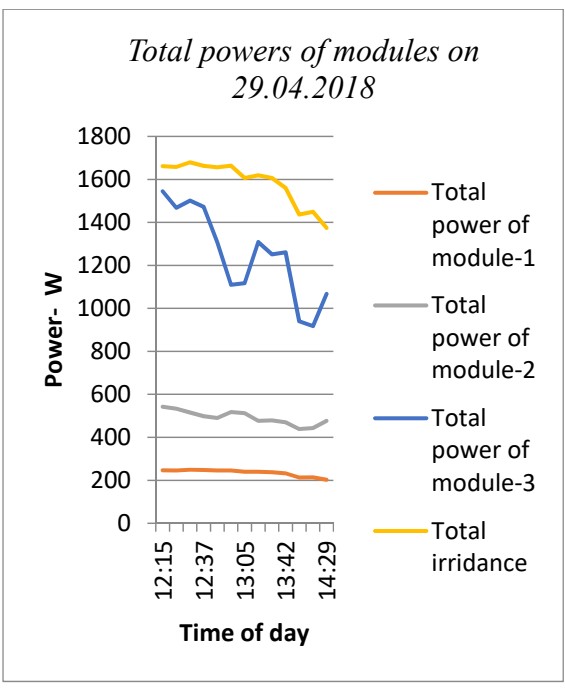

**Figure 16.** Total powers of modules on 29.04.2018.

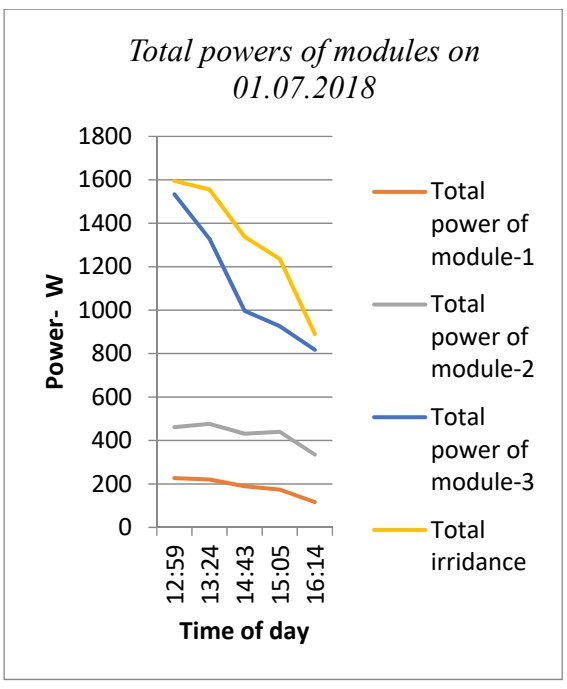

**Figure 17.** Total powers of moduleson 01.07.2018.

## 4. Results and Discussion

The total power in module-1 was nearly 200 W on each day because the experiment was conducted in clear sky conditions. In module-2 the total power was between 400–500 W on each day. On the other hand, the total power in module-3 was between 1600 W and 900 W on 29.04.2018 and was between 1600 W and 800 W on 01.07.2018. Both days had clear sky conditions but there was a decreased solar irradiance trend on 01.07.2018 (In Figures 16 and 17). In addition, the total efficiencies of the modules are seen in Figures 18 and 19. The total efficiency of module-1 was nearly 14% on each day. The total efficiency of module-2 was nearly 30% on 29.04.2018. However, the total efficiency in module-2 was

between 30% and % 40 on 01.07.2018. Finally, the total efficiency in module-3 was between nearly 60% and 90% on 29.04.2018. However, it was between nearly 75% and 90% on 01.07.2018. The change in total efficiencies in module-3 is clearly observed in Figures 18 and 19 on each day because the heat gain of air showed a sharp increase and decrease because of the sharp alteration of the input air temperature and the output air temperature.

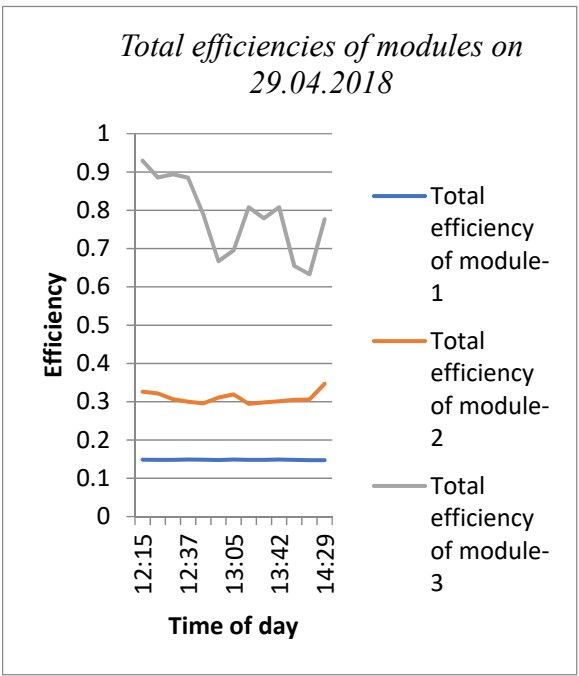

**Figure 18.** Totalefficiencies of modules on 29.04.2018.

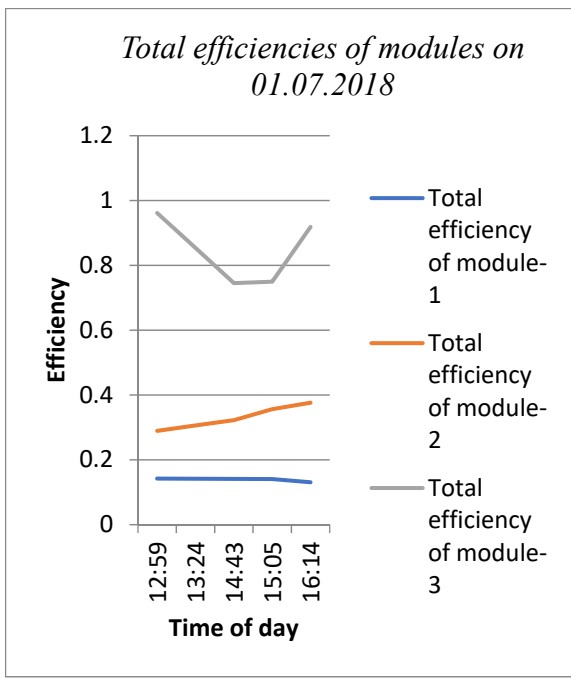

**Figure 19.** Total efficiencies of modules on 01.07.2018.

The efficiencies in module-1 were nearly 14% on each day. The total efficiencies in module-2 were between nearly 20% and 40% on each day. On the other hand, the module-3 total efficiencies were between nearly 60% and 90% on each day. While the average total efficiency was 84% on 29.04.2018,

on 01.07.2018 it was 78% in module-3. We can easily infer from the analyses above that the usage of water and air together is more efficient than a single usage of either for a PV/T system. The electrical efficiencies in module-2 and module-1 were close to each other on each day, but the electrical efficiency in module-2 on 01.07.2018 was closer to the efficiency of module-1 because the cooling was more effective on 01.07.2018. We can clearly see in Figures 6 and 7 that the body temperature in module-2 was less than module-1 on 01.07.2018. On the other hand, the electrical efficiency in module-3 was less than the other modules, because the glass cover had negative effects on the production of electrical energy. However, the total energy gain of module-3 was rather high. The usage of a black absorber plate, which was placed under the PV panel, increased the thermal energy gain but it had a negative effect on the cooling.

As a result, the thermal gain of working fluids was well high in module-3 and it could be used for several purposes by optimization. The photovoltaic-thermal system could be used where both thermal and electrical energies are needed; for instance, in hotels and dormitories. Furthermore, this system could be used in several drying processes and greenhouses. It could also be used in residential applications by optimization.

Having conducted this study in a natural ambient environment and under different atmospheric conditions, it turned out to be very difficult but certainly gave more genuine results.

## 5. Conclusions

According to climatic conditions from Konya Province in Turkey, this study shows that the electrical efficiency in the PV panel is very limited when it is used alone. However, according to the results of the study, an important amount of solar energy could be benefitted from the PV/T systems designed. Especially the usage of water and air together as a fluid in the photovoltaic thermal system is very suitable to gain energy. Adding that this system could be easily used for many purposes, it can be developed by various design applications. For example, various studies can be done on glass covers which have different optic features or different serpentine and absorber plate joints can also be investigated. With all the studies carried out and the qualities mentioned above, it is a high possibility that PV/T applications will take a major role in the construction of buildings in the near future, while solar applications will play an important role in all energy applications.

As a result, solar energy is a very useful kind of energy and it could be an answer for the energy problems that may appear in the future.

**Author Contributions:** M.A. conceptualization and supervision; I.Z.P. writing and draft preparation.

**Funding:** This research was supported by Marmara University Office of Scientific Research Projects. Project Number: 2017 FEN-C-DRP-070317-0111.

**Conflicts of Interest:** The authors declare no conflict of interest.

## Nomenclature

| | |
|---|---|
| $I_{sc}$ | Short circuit current (A) |
| $V_{oc}$ | Open circuit voltage (V) |
| $P_{th-m2}$ | Thermal power of module-2 (W) |
| $P_{th-m3}$ | Thermal power of module-3 (W) |
| $\dot{m}_{water}$ | Mass flow of water (kg/s) |
| $\dot{m}_{air}$ | Mass flow of air (kg/s) |
| $C_{p(air)}$ | Specific heat of air (j/kg°C) |
| $C_{p(water)}$ | Specific heat of water (j/kg°C) |

| $T_i$ | Inlet temperature (°C) |
|---|---|
| $T_o$ | Outlet temperature (°C) |
| $P_{tot\text{-}m2}$ | Total power of module-2 (W) |
| $P_{tot\text{-}m3}$ | Total power of module-3 (W) |
| $\mu_{th\text{-}m2}$ | Thermal efficiency of module-2 |
| $\mu_{th\text{-}m3}$ | Thermal efficiency of module-3 |
| $\mu_{o\text{-}m2}$ | Overall efficiency of module-2 |
| $\mu_{o\text{-}m3}$ | Overall efficiency of module-3 |
| $I_{tot}$ | Total irradiance (W) |
| $I_t$ | Solar Irradiance (W/m$^2$) |
| $A$ | PV module area (m$^2$) |
| $\mu_e$ | Electrical efficiency |
| $\lambda$ | Heat conduction coefficient (W/mK) |
| $T_i$ | Inlet temperature (°C) |
| $T_o$ | Outlet temperature (°C) |
| $T_{PV}$ | PV panel temperature (°C) |
| $T_a$ | Ambient temperature (°C) |

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
