# Peer review of "An Investigation on the Effect of the Total Efficiency of Water and Air Used Together as a Working Fluid in the Photovoltaic Thermal Systems"

_processes, doi:10.3390/pr7080516_

Round 1
Reviewer 1 Report
The article presents an interesting analysis of photovoltaic-thermal systems efficiency in Turkey, however the references presented by the authors are not totally update. It is recommended for the authors compare and evaluate they efficiencies and electrical powers analyses with recently and interesting similar researches, e.g.
Hussein A. Kazem, Evaluation and analysis of water-based photovoltaic/thermal (PV/T) system, Case Studies in Thermal Engineering, 2019; Volume 13, 100401, ISSN 2214-157X.
Jia, Yuting & Alva, Guruprasad & Fang, Guiyin, Development and applications of photovoltaic–thermal systems: A review, Renewable and Sustainable Energy Reviews, Elsevier, 2019; vol. 102(C), pages 249-265.
Alobaid, M., Hughes, B., O Connor, D. et al. (2 more authors), Improving thermal and electrical efficiency in photovoltaic thermal systems for sustainable cooling system integration, Journal of Sustainable Development of Energy, Water and Environment Systems, 2018; 6 (2), pp. 305-322, ISSN 1848-9257.
Marco Noro, Renato Lazzarin, Giacomo Bagarella, Advancements in Hybrid Photovoltaic-thermal Systems: Performance Evaluations and Applications, Energy Procedia, 2016; Volume 101, pp. 496-503, ISSN 1876-6102.
Dean, Jesse, McNutt, Peter, Lisell, Lars, Burch, Jay, Jones, Dennis, and Heinicke, David. Photovoltaic-Thermal New Technology Demonstration. United States: N. p., 2015.
Ewa Radziemska, Performance Analysis of a Photovoltaic-Thermal Integrated System, International Journal of Photoenergy, 2009; vol. 2009, Article ID 732093, p. 6.
It is recommended for the authors work on the english grammar of the paper, some figures and graphics titles distribution.
Best Regards
Author Response
Thank you so much your comments of our paper. We enhanced literature with updated researchs. Also we placed in the paper compare and evaluation with similar researches. Also we made some corrections in english grammer of the paper. Corrections and additions were marked colored in the paper.

Reviewer 2 Report
General comments:
The topic of the paper is very interesting, within the scope of the journal, and worthy of investigation. The originality of the work is acceptable. However, I suggest that authors should take into consideration the following comments before the manuscript can be published in Processes.
According to the mdpi guidelines, the first name should be entered before the last name. Is it like that?
What is the contribution of authors? Please add this part at the end of the manuscript.
Abstract:
Abstract introduces the readers with the subject of the investigation. It summarizes the scope and results of the research.
Introduction:
The introduction can be extended so that more reference works can be mentioned.
References should be numbered in order of appearance in the text.
The originality of the study should be more highlighted.
This part of the paper contains a lot of quotation marks. Is the text in quotes copied from other articles? These parts of the manuscript should be rewritten.
In a few places authors used the phrase “and his friends” (e.g. page 2 lines 58, 74, 80, 85; page 3 lines 91, 95). It is awkward to use such phrases in scientific paper. It should be changed.
System description:
A few sentences should be added at the beginning of the part 2.1 to familiarize the reader with the content of this section. Also, Figure 1 should be cited in this text, not in the subsection title.
Figures should be cited in the main text in order of appearance (currently Fig. 3 is cited before Fig. 2, Fig. 5 is cited before Fig. 4).
The quality of Figures should be improved. Figures 1 and 2 resolution looks low.
The title of subsection 2.2 should be changed.
What program has been used? (page 5 line 157)
NOMENCLATURE
There are missing units.
Moreover, I am not share if this is a good place for this part of the manuscript.
Analysis:
A few sentences should be added at the beginning of the part 3.1.
Page 8 line 219 (Eq. 14) – Please check.
In most Figures there are probably double spaces in the captions (before “of”).
In some Figures units are entered after “-“. In others, no. It should be standardized.
Figure 8, 12, 16 – “power“ should be changed on “Power”.
Figure 14, 18 – “efficiencies” should be changed on “Efficiencies”.
In some places authors used “Watt”. In others, “W”. It should be standardized.
Results and discussion:
The results are presented in section 3. Some parts of the paper should be moved. Otherwise, titles of sections should be changed.
Conclusions:
This section is poor and should be extended. Also, directions of future research should be added.
References:
References relevant to Processes should be added.
The style of references is not consistent with the guidelines.
Author Response
Responds to reviwer-2 attached

Round 2
Reviewer 1 Report
The authors made a progress in this new version, however there are still important issues that they must add and correct:
The authors didn’t include the reference that was mentioned in the last review comments:
· Jia, Yuting & Alva, Guruprasad & Fang, Guiyin, Development and applications of photovoltaic–thermal systems: A review, Renewable and Sustainable Energy Reviews, Elsevier, 2019; vol. 102(C), pages 249-265.
The above research work is from this year and summarize differents technologies of PV/T systems using different kinds of working fluids under a variety of environmental conditions. It will be interesting include a comparison between this results and author’s results.
The authors have gramatical issues in all the document that must be check, like the following examples:
Note: The next examples will refer to the final paper.pdf document lines
· The references includes the APA and journal formats togheter (e.g., Below, Bardhi et al. [6] in line 59, etc.). It is recommended rewrite the above
· Please change the text or be more specific in line 46 (Sirinivas and Jarayaj[2] ‘’were study’’ on the PV/T system......)
· The authors repeat references in followed paragraphs (e.g., from line 59 to line 62; from line 64 to line 67, etc.) that are related with the information of the same reference. It is recommended rewrite the above
· The authors continously repeat ‘’This present study’’, ‘’This study’’, ‘’On the other hand’’ several times (e.g., lines 102, 108, 109, 113, 123, 127, 128, 178, 317, 324, 334, 337, 341, 344), it is recommended use other synonyms in order to avoid repetitation.
· It is strongly recommended rewrite sentences like in line 265: ‘’We can see in all the figures (Fig. 8, Fig.9, Fig.10, Fig.11) below….’’ as ‘’ We can observe in Figures 8 to 11……’’ and line 337 (conclusion section): ‘’ This present study shows that the electrical efficiency in the PV panel is very limited…’’ as ‘’ According with climatic conditions from Konya Province in Turkey, this study shows that the electrical efficiency in the PV panel is very limited…’’
Please check the entire paper in order to rewrite sentences like the above examples.
· It is recommended rewrite the sentence in line 334 ‘’It is a sure that we will see more PV/T applications in our buildings in the future’’ or add the references which prove the above sentence.
· Correct the autor’s name Xu Li in reference 19.
Best Regards
Author Response
I attached response to comments
Best regards,

Reviewer 2 Report
You have taken into account a lot of my comments. However, I still have some doubts about several things:
- Why the name of the third author has been removed?
- The letters “a” and “b” are left behind the names of the authors. Is it intentional? (p. 1, line 5)
- The text of the paper is very poorly formatted (different line spacing, double spaces, no spaces between words, etc.). Please correct the formatting of the manuscript.
- I also have doubts about the nomenclature. I suggested you to add units, but you didn’t do it.
- What’s more, in my opinion Figures 6-19 are the same as in the previous version of the paper. My comments were not included.
Author Response

(The authors gave the same response as above.)

Round 3
Reviewer 1 Report
It is recommended to check the english grammar in all the document and rewrite some sentences with the words ''figures'' inside the parenthesis, e.g. line 341 in the word document:
''We can see the decrease trend in the total powers of the modules (Figure 16, Figure 17)'', it can rewrite as ''In Figures 16 and 17 we can see the decrease trend in the total powers of the modules''
Best Regards and thank you for your time
Author Response
I attached respont reviewer
Best regards,
